

# A new gnathosaurine (Pterosauria, Archaeopterodactyloidea) from the Late Jurassic of Portugal

Alexandra E. Fernandes[1,2,3], Victor Beccari[2,3], Alexander W. A. Kellner[4] and Octávio Mateus[3,5]

[1] Department of Earth and Environmental Sciences, Ludwig-Maximilians-Universität München, Munich, Bayern, Germany
[2] SNSB, Bayerische Staatssammlung für Paläontologie und Geologie, Munich, Germany
[3] Museu da Lourinhã, Lourinhã, Portugal
[4] Departamento de Geologia e Paleontologia, Laboratório de Sistemática e Tafonomia de Vertebrados Fósseis (LAPUG), Museu Nacional, Rio de Janeiro, Brazil
[5] GEOBIOTEC, Department of Earth Sciences, Universidade Nova de Lisboa, Caparica, Portugal

## ABSTRACT

An incomplete, yet remarkably-sized dentated rostrum and associated partial cervical vertebrae of a pterosaur (ML 2554) were recently discovered from the Late Jurassic (Late Kimmeridgian-Early Tithonian) Lourinhã Formation of Praia do Caniçal, of central west Portugal. This specimen exhibits features such as a spatulated anterior expansion of the rostrum, robust comb-like dentition, and pronounced rims of the tooth alveoli, indicating gnathosaurine affinities. Based on its further unique tooth and dentary morphology, a new genus and species, *Lusognathus almadrava* gen. et spec. nov., is proposed, making this the first named pterosaur species found within Portugal. The presence of this taxon adds yet another element to the fluvio-deltaic lagoonal environment that has been suggested as representative of the Lourinhã Formation in the Late Jurassic, further contributing to the diversity and distribution of gnathosaurines worldwide.

## INTRODUCTION

The known global distribution and diversity of pterosaurs reinforces their success as a group, as they are found in all continents including Antarctica (*Barrett et al., 2008*; *Kellner et al., 2019a*), and yet their relatively sparse fossil record and often incomplete preservation (particularly when outside of Lagerstätten environments) can pose a challenge for further understanding their paleobiology, when compared with other vertebrates. Accordingly, the Jurassic of Portugal is a very productive and taxonomically diverse period concerning vertebrate fossils, especially for plesiosaurs (*e.g.*, *Puértolas-Pascual et al., 2021*), ichthyosaurs (*e.g.*, *Castanhinha & Mateus, 2007*), mosasaurs (*e.g.*, *Castanhinha & Mateus, 2007*), dinosaurs (*e.g.*, *Rauhut, 2001*; *Antunes & Mateus, 2003*; *Malafaia et al., 2010*; *Mocho et al., 2016*; *Rotatori, Moreno-Azanza & Mateus, 2020*), turtles (*e.g.*, *Pérez-García & Ortega, 2011*), crocodylomorphs (*e.g.*, *Guillaume et al., 2020*), and mammals (*e.g.*, *Krebs, 1991*). However, despite this abundance, up to now, pterosaur material recovered from this

Corresponding author
Octávio Mateus, omateus@fct.unl.pt

deposit has been restricted to scant and often fragmentary isolated bones and teeth, hindering any confident taxonomic assignments. This is likely due to the physical bone fragility of pterosaurs, making their remains particularly susceptible to deterrent or destructive fossilization factors such as carcass scavenging or later taphonomic duress (*Dean, Mannion & Butler, 2016*).

On a worldwide level, whereas ctenochasmatids occur with some regularity throughout the fossil record landscape for pterosaurs from the Late Jurassic to the Early Cretaceous, gnathosaurines are significantly more rare (*Barrett et al., 2008*). The currently-known gnathosaurine temporal range spans from the Late Jurassic to the Early Cretaceous, and although their distribution has so far extended throughout Europe, Asia, and South America, most of the known occurrences are attributed to isolated teeth, which have been reported from England, Portugal, Morocco, Chile, Uruguay, China, and Japan (*Barrett et al., 2008*; *Howse & Milner, 1995*; *Knoll, 2000*; *Martill et al., 2006*; *Perea et al., 2018*; *Soto et al., 2021*; *Sweetman & Martill, 2010*; *Zhou et al., 2016*; *Dong, 1982*; *Unwin, Lü & Bakhurina, 2000*).

The fossil material herein described now introduces a new taxon to the fluvio-deltaic lagoonal environment that has been suggested as representative of Lourinhã Formation in the Late Jurassic. The material is housed at the Museu da Lourinhã, in Lourinhã, Portugal, under the collection number ML 2554.

## History and record of pterosaur discoveries in portugal

The presence of pterosaurs in Portugal (Fig. 1) was first reported by Lapparent and Zbyszewski (1957: p. 58), the material consisting of four small, elongated vertebrae from the Campanian/Maastrichtian of Viso, which were initially identified as being similar to *Dimorphodon* or *Rhamphorhynchus*, and later ascribed to a maniraptoran theropod dinosaur by *Galton (1994)*. The author also claimed in the same work that a pterodactyloid cervical vertebra from the middle portion of the neck, likely the fifth (formerly MSGP N X 213, and now belonging to the Museu Geológico (MG) in Lisbon, Portugal), discovered in the Lower Cretaceous Barremian of Serra Tiago dos Velhos was actually the first record of a pterosaur from Portugal, which he ascribed to cf. *Ornithocheirus*.

In 1968, Kühne initially attributed remains from the Kimmeridgian beds of the Guimarota mine (teeth and a terminal manual phalanx) to two different groups of pterosaurs: aff. *Rhamphorhynchus* sp. and *Pterodactylus* sp. (*Kühne, 1968*). Later, this material, in conjunction with other isolated fragmentary remains composed of a left scapula, wing? elements, and other appendicular elements could only be referred to the Pterodactyloidea, family *incertae sedis* (*Thulborn, 1973*; *Wiechmann & Gloy, 2000*). Guimarota has also yielded over 300 isolated pterosaur teeth, attributed to *Rhamphorhynchus* sp. and to the Pterodactyloidea (*Wiechmann & Gloy, 2000*). The fossils from Guimarota are also deposited in the Museu Geológico, in Lisbon, Portugal.

Other isolated teeth comprise the vast majority of pterosaur fossils from Portugal, and were recovered from the Areia do Mastro locality of the lower Barremian Papo Seco Formation of Cabo Espichel. They were assigned to the Ornithocheiridae and to the Ctenochasmatoidea (*Figueiredo et al., 2020*, *2022*). From the Andrés fossil site of the

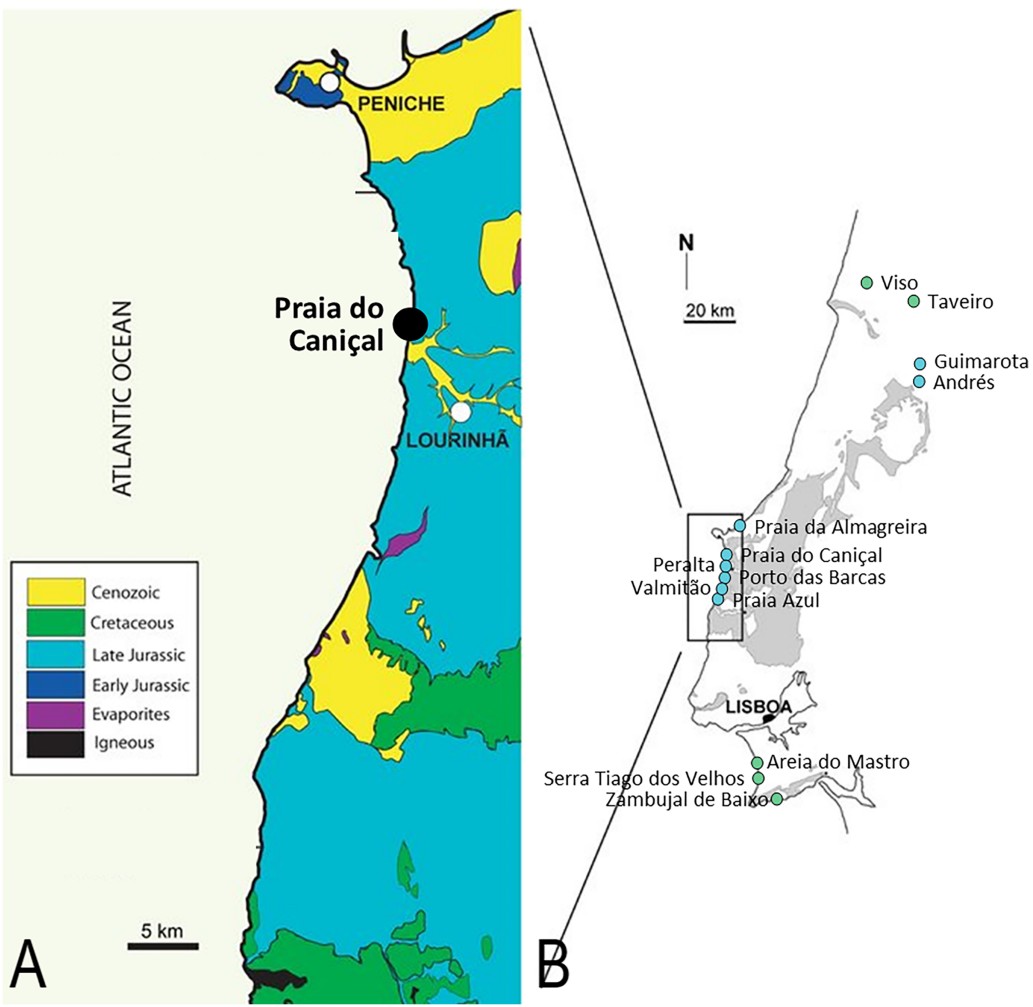

**Figure 1 Pterosaur localities of Portugal.** Locality of ML 2554 of Praia do Caniçal (A) and other known pterosaur localities of Portugal (B) (image modified from *Mateus, Dinis & Cunha, 2017*)

Kimmeridgian Alcobaça Formation, isolated teeth were also attributed to Pterosauria indet. *Malafaia et al. (2010)*, based on their needle-like morphology (and similar in morphology to the *Rhamphorhynchus* sp. Guimarota teeth). Isolated teeth were also recovered in the area of Valmitão in Lourinhã (*Guillaume et al., 2020*), and teeth assigned to *Gnathosaurus* sp. were reported from the Sobral Formation (late Kimmeridgian-early Tithonian) locality of Praia Azul in Torres Vedras (*Bertozzo et al., 2021*).

Some fragmentary appendicular bones have been tentatively attributed to pterosaurs, including a large femur attributed to Dsungaripteroidea, from the locality Praia da Almagreira, Peniche (*Bertozzo et al., 2021*) of the upper Kimmeridgian/lower Tithonian. Some indeterminate pterosaur bone fragments were also reported from the Upper Cretaceous sediments of Taveiro, without any particular taxonomic assignment (*Antunes & Pais, 1978*).

About 400 *Pteraichnus* tracks have been collected from the Kimmeridgian/Tithonian Amoreira-Porto Novo Member locality of Peralta, which may correspond to the new taxon described in this work, although a more decisive attribution of these ichnofossils is beyond the scope of the current contribution. Trackways of *Pteraichnus* sp. (*Mateus & Milàn, 2010*) are also reported from the Upper Jurassic (Kimmeridgian) Zambujal de Baixo locality of the Azóia Formation in Sesimbra and Porto das Barcas in Lourinhã (late Kimmeridgian-early Tithonian).

Overwhelmingly, the vast majority of previous pterosaur specimens from Portugal have been assigned to *Rhamphorynchus* sp. However, this was likely a historically-generalized temporal attribution (rather than strictly based on morphology), and the known Portuguese material needs to be revised, especially considering that the range of paleobiogeographical variability has substantially grown since the time of these original attributions. In fact, no definitive non-tooth skeletal material of *Rhamphorynchus* sp. has ever even been reported in Portugal to date, which further warrants the potential for misnomers.

Here we present and describe a new specimen found by Filipe Vieira in November 2018 at Praia do Caniçal, in the municipality of Lourinhã in central west Portugal. Further excavation efforts realized in March 2019 by members of the Museu da Lourinhã collected more material from this same specimen, which is the first pterosaur taxon named from this country.

## Geographical and geological setting

At Praia do Caniçal the outcropping rocks are of the Upper Jurassic of the Lourinhã Formation, namely the middle member of this unit: the Praia Azul Member. The Lourinhã Fm. displays a succession of alternated terrestrial mudstones and sandstones, some paleosol levels, and three transgressive bioclastic brackish layers. The formation is mostly continental, as evidenced by the presence of terrestrial fauna such as lissamphibians, non-aquatic mammals, and dinosaurs. The paleoclimate was arid (*Myers et al., 2012*) and the paleolatitude, derived from http://paleolatitude.org (*van Hinsbergen et al., 2015*), is estimated as 28°–29° North.

The Praia Azul Member is regarded to have been deposited near-shore, due to sea transgression. It is characterized by marls, mudstones, and sandstones, and is composed of three conspicuous carbonate levels (with the lower and upper levels used as the lithostratigraphic boundaries) and can be traced over distances of 20 km. The unit was mainly deposited by meandering fluvial systems flowing in a low-lying coastal plain, connected with transitional systems like deltas, sandy bay shorelines, and brackish lagoons. The brackish faunas of the shelly layers indicate short-time marine incursions (see *Manuppella et al., 1999*; *Hill, 1989*; *Martinius & Gowland, 2011*; *Myers et al., 2012*; *Taylor et al., 2014*; *Mateus, Dinis & Cunha, 2017*). So far, all studies agree to an Upper Kimmeridgian-Lower Tithonian age of those layers, which is also supported by an only non-biostratigraphic inferred date, provided by strontium isotopes by *Schneider, Fürsich & Werner (2009)*. More precisely, the second and middle transgressive layer is at the Kimmeridgian-Tithonian boundary, which is currently at 149.2 Ma.

The Praia Azul Mb. is composed of three transgressive carbonate layers with different faunal assemblages: the lower level is categorized by *Isognomon lusitanica*, *Eomiodon securiformes*, *Arcomytilus morrisii*, dinosaur tracks, and ostreids (outcropping north of Caniçal, below the Paimogo fort) (*Ribeiro et al., 2014*; *Mateus, Dinis & Cunha, 2017*). The middle level comprises *Jurassicorbula edwarsi*, *Isognomon lusitanica*, *Nerinea*, coprolites, fish, *Eomiodon securiformes*, echinoids (outcropping at the base of Praia do Caniçal), while the upper level fauna consists of *Jurassicorbula edwarsi*, abundant ostreids, pleurosternidae turtles, and *Isognomon lusitanica* (outcropping at the top of Praia do Caniçal) (*Mateus, Dinis & Cunha, 2017*). The member is known to bear abundant dinosaur remains, including bones, eggs, and tracks (*Ribeiro et al., 2014*; *Mateus et al., 2011*), and also crocodylomorphs (*Young et al., 2014*), turtles (*Pérez-Garcí, 2015*; *Pérez-García & Ortega, 2022*), pterosaurs (*Bertozzo et al., 2021*), mammals, lissamphibians, and lepidosaurs (*Mateus, Dinis & Cunha, 2017*; *Guillaume et al., 2023*).

The specimen studied here was excavated from a stratigraphic layer at modern intertidal sea level, below the sand level of the beach itself, which is only exposed at low tide during the winter months (when sand is absent), which complicated its extraction. The fossil was preserved in a reddish micaceous fine sandstone, about 4 m above the second (and middle) transgressive layer, and below the upper transgressive layer. The age of ML 2554 is thus estimated to be about 149 Ma (*Schneider, Fürsich & Werner, 2009*).

## MATERIALS AND METHODS

The preparation of the specimen was done mechanically, using PaleoTools Micro Jacks and a variety of manual tools, all performed at the Museu da Lourinhã in Lourinhã, Portugal. All bone surfaces were consolidated with 5% Paraloid B-72 diluted in acetone, and any breaks or deep fissures were glued and reinforced with 20% or 50% Paraloid B-72, as required.

CT scans were performed in Leiria, Portugal, with a microfocus CT system GE VtomeX M 240. Scan images were segmented and assembled utilizing Avizo and Meshlab softwares (GE Sensing & Inspection Technologies GmbH., Wunstorf, Germany). This resulted in four stacks of DICOM (.dicom) images (detailed information available at Morphobank Project 3968) (*O'Leary & Kaufman, 2012*). The segmentation of the complete specimen was done using manual selection slice-by-slice in the software Avizo v9.1 (Thermo Fisher, Waltham, MA, USA). All meshes were exported as Wavefrontfiles (.obj) and treated in the open-source software Blender v3.4. All meshes were smoothed for rendering using the Smooth Laplacian modifier (Lambda factor = 1 and 10 repeats). Measurements were taken both directly from the physical specimen and digitally in Blender.

Phylogenetic analysis was conducted using TNT version 1.5 (*Goloboff, Farris & Nixon, 2008*; *Goloboff & Catalano, 2016*) using the matrix by *Andres (2021)*, augmented by the additional Portuguese specimen described here (represented in the analysis by "Lusognathus_almadrava"). A basic traditional tree-search analysis was conducted with 1,000 random addition sequence replicates. Due to the focus on the interrelationships of the new specimen and ctenochasmatids, the resulting cladogram has been simplified in the

software Adobe Illustrator. The complete topology can be found in the supplementary data set.

All specimen CT data, digital material, and phylogenetic matrix files are available at Morphobank Project 3968 (*O'Leary & Kaufman, 2012*).

## Nomenclatural acts

The electronic edition of this article conforms to the requirements of the amended International Code of Zoological Nomenclature, and hence the new names contained herein are available under that Code from the electronic edition of this article. This published work and the nomenclatural acts it contains have been registered in ZooBank, the online registration system for the ICZN. The ZooBank LSIDs (Life Science Identifiers) can be resolved and the associated information viewed through any standard web browser by appending the LSID to the prefix "http://zoobank.org/". The LSID for this publication is: urn:lsid:zoobank.org:pub:54B6D020-612A-4E23-ADEC-2CDE383B5C3B.

The electronic edition of this work was published in a journal with an ISSN, and has been archived and is available from the following digital repositories: PubMed Central and LOCKSS.

## RESULTS

### Systematic Paleontology

**PTEROSAURIA** *Owen, 1842*

**PTERODACTYLOIDEA** Plieninger, 1901

**ARCHAEOPTERODACTYLOIDEA** *Kellner, 2001*

**CTENOCHASMATIDAE** *Nopsca, 1928 sensu Unwin, 2003*

**GNATHOSAURINAE** *Nopsca, 1928 sensu Unwin, 2002*

*Lusognathus*, gen. nov.

*L. almadrava*, sp. nov.

Etymology: *Lusognathus* is derived from the Latin "Luso", after the prefix used for referencing things relating to Lusitania (the former name for the area of Portugal in Roman times) and "gnathus" meaning "jaw"; an "almadrava" is the name of a traditional Portuguese fishing trap for catching seafood.

Holotype: ML 2554, comprised of a fragment of the anterior part of a premaxillary rostrum, a fragment of a maxillary toothrow, two isolated fragmentary teeth, and three (or four) partial fragments of cervical vertebrae.

Locality and Horizon: Praia de Caniçal, county of Lourinhã, Portugal. Lourinhã Formation, late Kimmeridgian-early Tithonian, about 149.2 Ma (*Schneider, Fürsich & Werner, 2009*).

Diagnosis: A gnathosaurine pterosaur with the following combination of characters: a rounded-triangular anterior expansion of the premaxilla, constriction of the maxilla

directly posterior to the spatulate anterior expansion, robust laterally-projected teeth (spaced at 1.3 teeth per cm) with a subcircular to oval cross-section, and posterior teeth projected anterolaterally.

### Specimen Description

Specimen ML 2554 is composed of a well-preserved three-dimensional maxillopremaxillary rostrum fragment with three(?) associated fragmentary mid-cervical vertebrae (two of which, although preserved in articulation, are too fragmentary to provide detailed anatomical information). The rostral fragments are divided between three blocks with only one side exposed. Two of the blocks connect and comprise the anteriormost premaxillary portion of the rostrum (Fig. 2), visible in ventral view, and the third block houses a more posterior maxillary section (of unknown exact position along the rostrum). No sutures are visible anywhere along the rostrum, although two longitudinal grooves are visible, running parallel to the toothrow (difficult to distinguish due to slight dorsoventral taphonomic compression, likely from lithostatic pressure). For precise measurements of each individual element, see Tables 1 and 2.

The anteriormost portion of the premaxilla is somewhat eroded. The increasingly anteromedial direction of the preserved roots of the anterior-most teeth indicate that the termination of the rostrum was not far beyond what is actually preserved (Figs. 2 and 3). In dorsoventral view, the rostrum begins to expand anterolaterally at approximately 28.0 mm from the anteriormost preserved edge, into a spatulate A-line or rounded-triangle shape, with the widest point terminating at the alveolar collar of the largest anterior tooth. Posterior to this point of expansion, and with no sharp demarcation (but still a distinct constriction) of the rostrum, a gradual posterolateral expansion also begins towards its posterior edge, but with a more gradual degree of increase than the anterior spatulated expansion.

The third rostrum fragment, which contains the posteriormost portion of the preserved maxilla, still preserves tooth alveoli, indicating that it was likely positioned anterior to the nasoantorbital fenestra. However, it should be noted that the actual percentage of the overall rostrum that is represented by this specimen cannot be determined.

There are twenty-nine teeth preserved in the rostrum, although in varying conditions, with one almost complete. Two additional isolated fragmentary teeth were also recovered from the surrounding sediment during preparation. Under these constraints, it is not possible to ascertain an exact overall tooth count for the taxon, although its tooth density can be calculated at approximately 1.3 teeth per cm (per side). The teeth along the premaxilla and maxilla exhibit a clear thecodont dentition, with each tooth located in a single alveolus which forms a raised collar of bone at the base of each tooth. The rostrum exhibits at least sixteen well-preserved alveoli on each side. The depressions between the alveoli gives the rostrum a crenellated appearance in dorsal and ventral views. The teeth are robust, reaching up to 5 mm in width, show a slight anterolateral tilt, and become increasingly inclined laterally and anteriorly. From the anterior to posterior end of the toothrow, the teeth also diminish in size and begin to slightly recurve posteriorly. The distance between individual teeth is greater than the tooth width, with the exception of

| Table 1 Measurements of *Lusognathus almadrava* ML 2554. | | |
|---|---|---|
| **Element** | **Left (mm)** | **Right (mm)** |
| Alveoli count | 11 | 17 |
| Tooth 3 root length | 9.1 | – |
| Tooth 3 diameter | 3.1 | – |
| Tooth 4 length | 23.5 | – |
| Tooth 4 root length | 12.1 | – |
| Tooth 4 diameter | 5.6 | – |
| Tooth 5 length | 20.4 | – |
| Tooth 5 root length | 8.1 | 8.3 |
| Tooth 5 diameter | 3.9 | 3.8 |
| Tooth 6 root length | 7.6 | 6.1 |
| Tooth 6 diameter | 3.7 | 2.7 |
| Tooth 7 root length | 4.9 | 3.2 |
| Tooth 7 diameter | 3.2 | 3.2 |
| Tooth 8 root length | 6.8 | 5.6 |
| Tooth 8 diameter | 3.4 | 2.8 |
| Tooth 9 root length | – | 4.2 |
| Tooth 9 diameter | – | 2.6 |
| Tooth 10 root length | 6.8 | 4.9 |
| Tooth 10 diameter | 3.5 | 2.9 |
| Tooth 11 root length | – | 4.5 |
| Tooth 11 diameter | 3.6 | 2.9 |
| Tooth 12 root length | 4.8 | 3* |
| Tooth 12 diameter | 2.8 | 3.3 |
| Tooth 13 root length | 5 | 5.1 |
| Tooth 13 diameter | 1.9 | 2.7 |
| Tooth 14 root length | 4.8 | 4.9 |
| Tooth 14 diameter | 2.7 | 3.1 |
| Tooth 15 root length | – | – |
| Tooth 15 diameter | 2.1 | – |
| Tooth 16 root length | 5.1 | – |
| Tooth 16 diameter | 2.5 | – |
| Tooth 17 root length | 5.2 | – |
| Tooth 17 diameter | 1.7 | – |
| Tooth 18 root length | 5.8 | – |
| Tooth 18 diameter | 2.6 | – |
| Average root length | 6.6 | |
| Average tooth diameter | 3.5 | |

the largest preserved tooth at the anterior end. In the transverse view, the gradation of the tooth row is not perfectly horizontal, and shifts slightly dorsally at the posterior region.

The preserved teeth have straight crowns, tapering towards the apex. They are subcircular to oval in cross-section, with a slight compression on their anteroventral and

| Table 2 Measurements of *Lusognathus almadrava* ML 2554. | | |
|---|---|---|
| Element | Left (mm) | Right (mm) |
| Maxilla length | 127.8 | |
| Maxilla maximum width | 40.5 | |
| Mid-maxilla width | 25.5 | |
| Maxilla height | 7.5 | |
| Mid-premaxilla width | 8.9 | |
| Distance Tooth (T) 3 to 4 | – | 3.5 |
| Distance T4-5 | – | 3.2 |
| Distance T5-6 | – | 4 |
| Distance T6-7 | 3.6 | 3.9 |
| Distance T7-8 | 5.2 | 4.9 |
| Distance T8-9 | 4 | – |
| Distance T9-10 | 5.3 | – |
| Distance T10-11 | 5.8 | 4.7 |
| Distance T11-12 | 5.4 | – |
| Distance T12-13 | 6.4 | 6 |
| Distance T13-14 | 3.9 | 4.9 |
| Distance T14-15 | 6 | 6.9 |
| Distance T15-16 | 7.1 | 6.7 |
| Distance T16-17 | – | 6.9 |
| Distance T17-18 | – | 5.5 |
| Distance T18-19 | – | 6.3 |
| Cervical vertebra centrum length | 35.7 | |
| Cervical vertebra preserved length | 49.9 | |
| Cervical vertebra centrum width | 20.9 | |
| Cervical vertebra neural spine height | 8.9 | |
| Cervical vertebra preserved height | 29.6 | |

**Note:**
Tooth distances were measured from the distal margin of the tooth to the mesial margin of the following tooth.

posterodorsal sides, resulting in keel-like attenuations running longitudinally. The tooth enamel is completely smooth and lacks any ornamentation (Fig. 3A). Where teeth have been broken, pulp cavities are readily visible.

The best-preserved vertebral fragment consists of the incomplete anterior portion of a cervical centrum (Fig. 4), exhibiting the prezygapophyseal and cotylar region. It exhibits slight taphonomic distortion but preserves its three-dimensionality. The neural spine is low. The neural canal, flanked by two depressions on either side, is visible as a deep divot centered on the anterior end. The bifid prezygapophyes project anterolaterally, and their articular faces turn anteromedially.

There are three (possibly four) remaining vertebral fragments. One block contains one or two fragmentary vertebrae preserved in the sediment, and are potentially the posterior end of the atlas-axis complex articulated with the first cervical vertebra. Of the two other articulated fragments, one appears to be an elongated posterior part of a centrum, and the

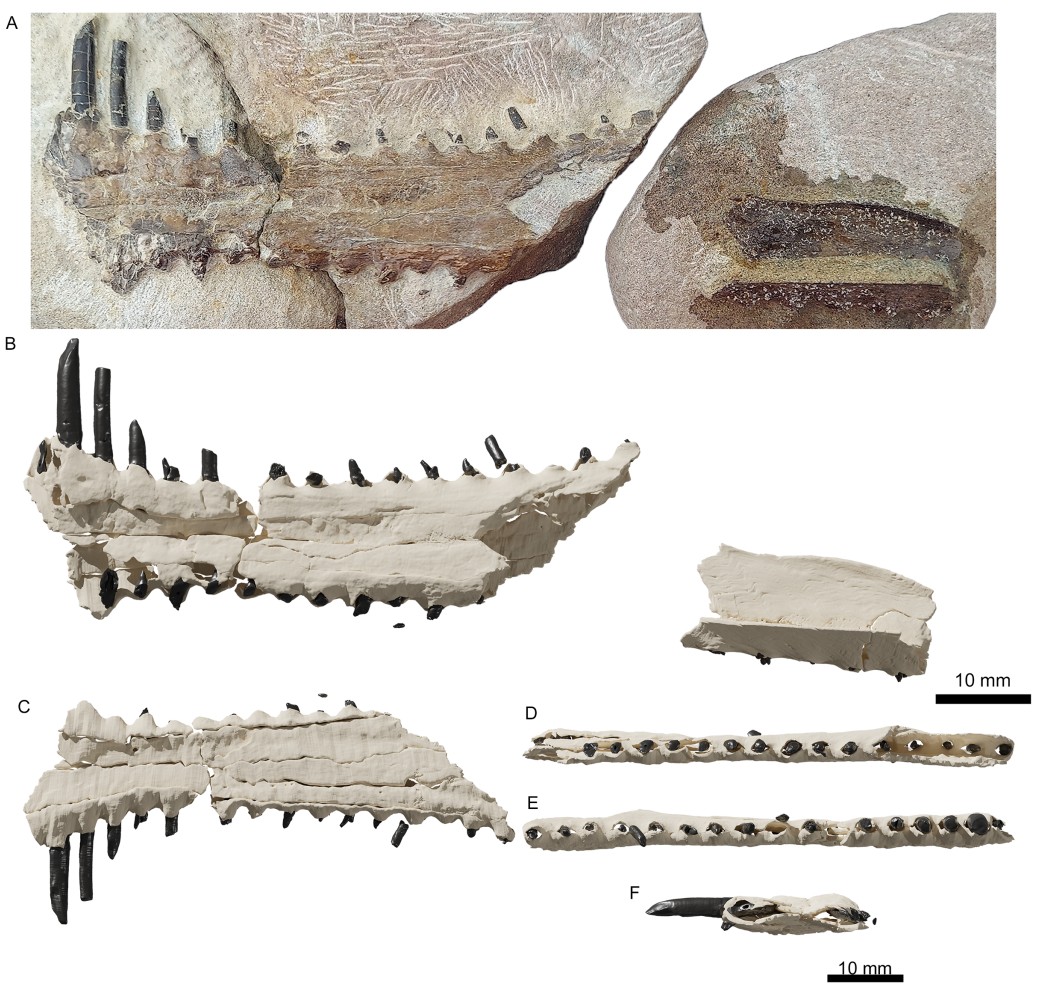

**Figure 2** **The upper jaw of the holotype (ML 2554) of** *Lusognathus almadrava* **gen. et sp. nov. (ML 2554).** (A) Photography of both jaw fragments; (B) CT-scan reconstruction of the jaw in dorsal view, showing minimum possible preserved length, in dorsal view; (C) ventral view; (D) left lateral view; (E) right lateral view; (F) anterior view.

other an anterior prezygapophyseal region of another vertebra (with a pronounced neural spine). This material, however, is too damaged to provide much information. Judging by the height of the neural spine (when compared with the most complete vertebra), this spine appears slightly taller and more pronounced.

No sign of a premaxillary crest could be distinguished along the rostrum. However, since premaxillary crest in gnathosaurines usually begin to take shape in a more posterior position (*e.g.*, *Gnathosaurus*), and the part of the rostrum preserved in *Lusognathus almadrava* is most likely well anterior to the nasoantorbital fenestra, we cannot be certain if this taxon would in fact bear one.

## Phylogenetic Analysis

Phylogenetic analysis was conducted analyzing the evolutionary relationships of *Lusognathus almadrava* and other pterosaurs using the data matrix of *Andres (2021)*. The analysis resulted in a single most parsimonious tree (Fig. 5). *Lusognathus almadrava*

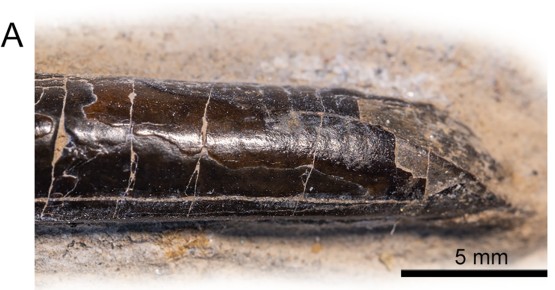

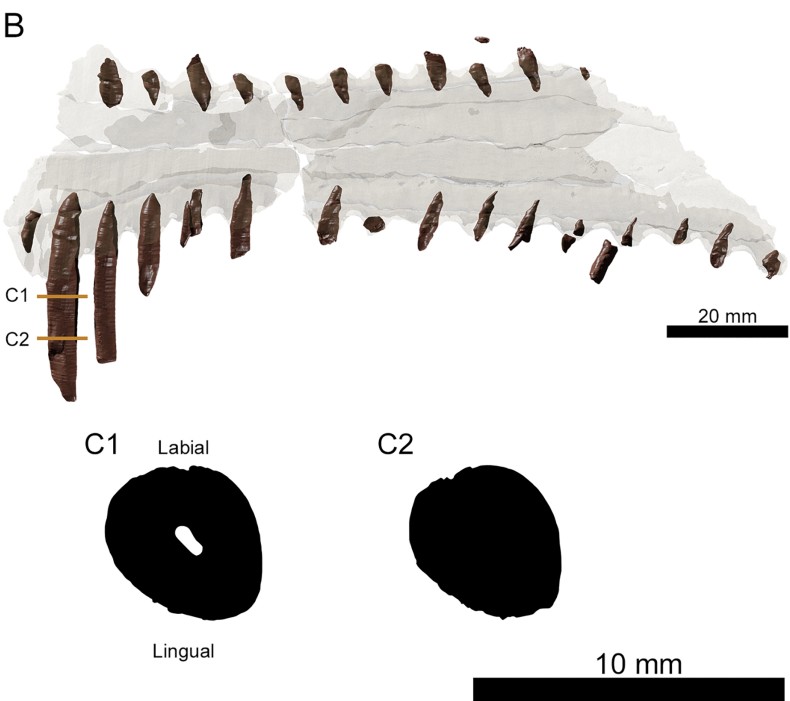

**Figure 3 Tooth morphology of *Lusognathus almadrava* gen. et sp. nov. holotype (ML 2554).** (A) Photography of the best-preserved tooth in labial view; (B) transparent 3D representation of the upper jaw in ventral view, showing the insertion of teeth, with cross-section cuts (C1 and C2) of the best-preserved tooth.

was retrieved as the sister taxon of *Gnathosaurus*, a relationship supported by a triangular lateral expansion of the anterior end of the rostrum (character 59.1). The Gnathosaurinae was phylogenetically defined as the least inclusive group including *Gnathosaurus subulatus von Meyer, 1834* and *Huanhepterus quingyangensis Dong, 1982* (*Andres, 2021*), and hence, *Lusognathus almadrava* falls within the Gnathosaurinae.

**Discussion/Taxonomic Assignment**

Typically, the defining feature of the Ctenochasmatidae (*Nopsca, 1928*) has been the tip of the rostrum being dorsoventrally depressed and rounded, whereas the designation for the gnathosaurines therein is the significant lateral expansion of the anterior end of the

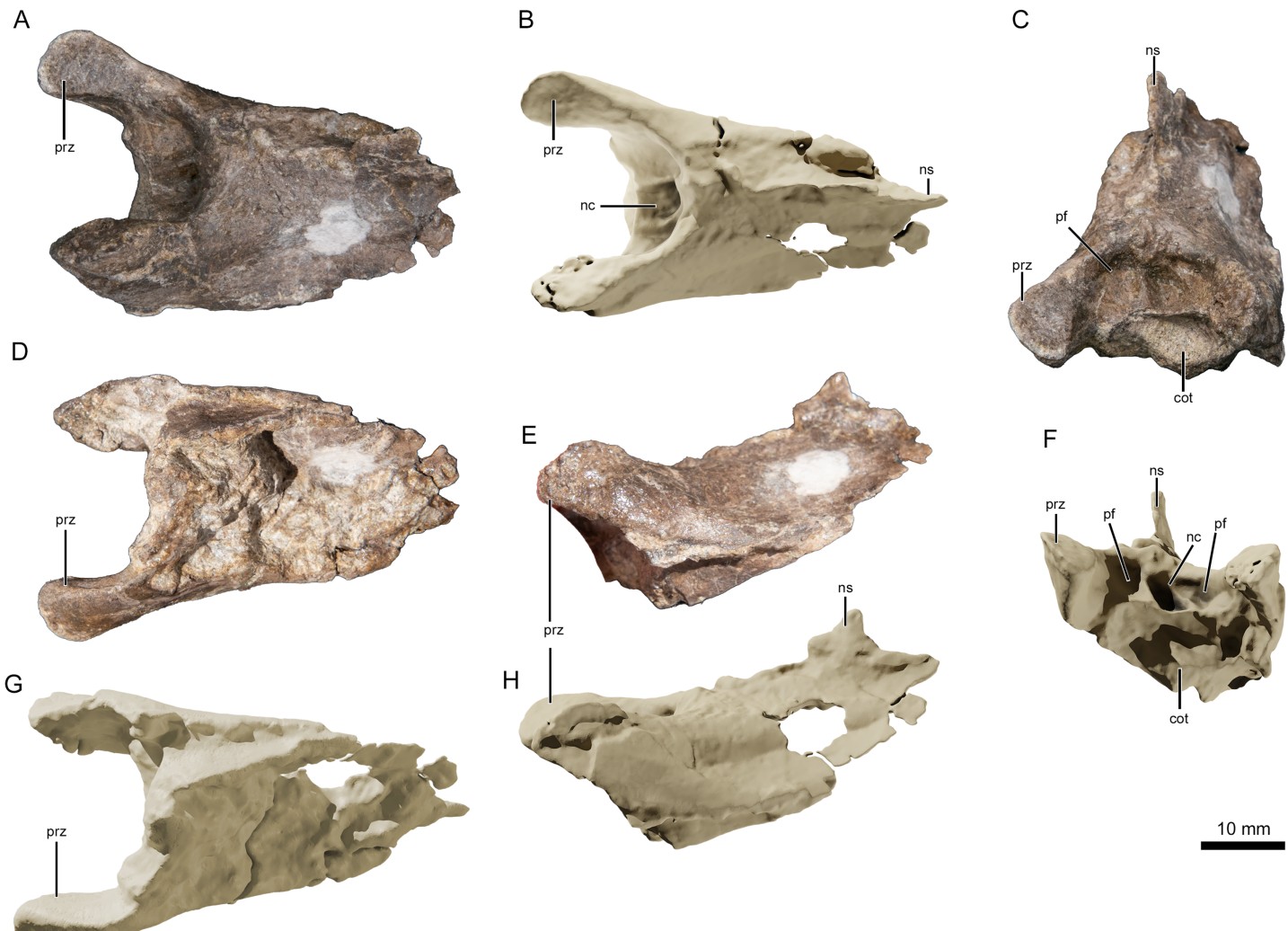

**Figure 4 Photographs and virtual three-dimensional renderings of the mid-cervical vertebra of *Lusognathus almadrava* gen. et sp. nov. holotype (ML 2554).** (A and B) Photograph and 3D rendering in dorsal view; (C and F) photograph and 3D rendering in anterior view; (D and G) photograph and 3D rendering in ventral view; (E and H) photograph and 3D rendering in left lateral view. Abbreviations: cot, cotyle; nc, neural canal; ns, neural spine; pf, pneumatic foramen; prz, prezygapophysis.                

rostra (*Unwin, 2002*). However, this expanded feature is not exclusive to gnathosaurines alone, as is also presents to a degree in anhanguerids (*e.g., Kellner, 2003*), istiodactylids (*e.g., Lü, Xu & Ji, 2008*), and ornithocheirids (*e.g., Unwin, 2002*, but see *Rodrigues & Kellner, 2013* for the status of Ornithocheiridae). What does make this feature unique for gnathosaurs (beyond the aforementioned dorsoventral rostral composition of the larger Ctenochasmatidae) is the pronounced extent and variation of the spatula, *e.g.*, the spoon-shaped spatulas of *Tucuadactylus luciae* (*Soto et al., 2021*), *Gnathosaurus macrurus* (*Howse & Milner, 1995*), and *Gnathosaurus subulatus* (Meyer, 1833) (Fig. 6A), and the more bulbously circular end of *Plataleorhynchus streptophorodon* (*Howse & Milner, 1995*) (Fig. 6C). The extent of the diversity of these shapes has only been revealed as more specimens have been discovered (*e.g., Soto et al., 2021*). Specimen ML 2554 sustained

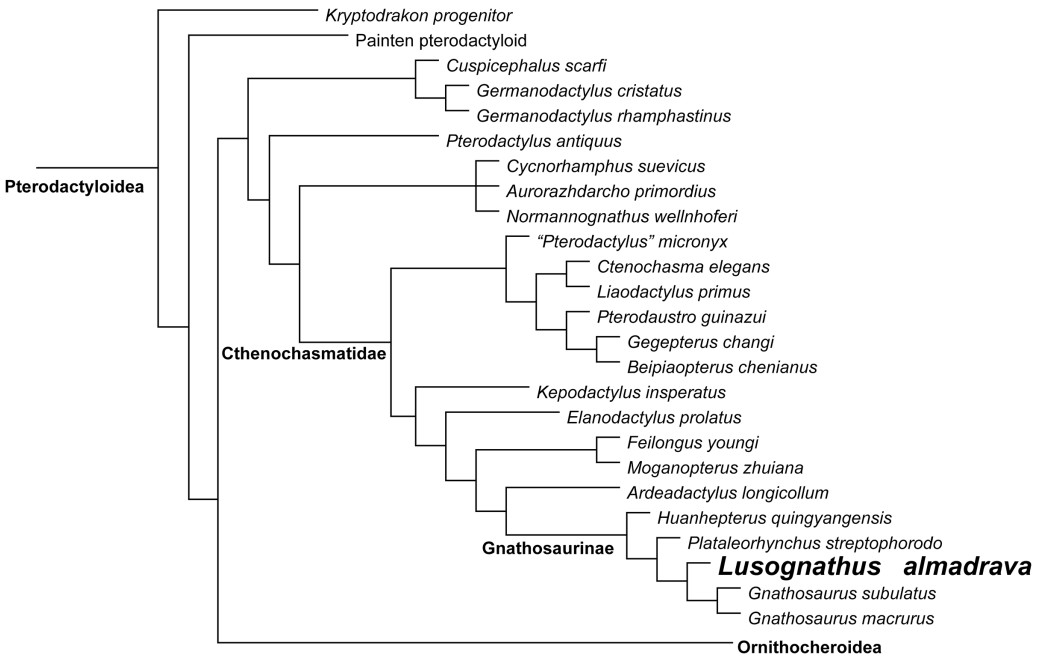

**Figure 5 Phylogenetic results.** Simplified tree of the Pterodactyloidea, showing the relationship of *Lusognathus almadrava* gen. et sp. nov., and gnathosaurines, after the data matrix by *Andres (2021)*. Non-pterodactyloid pterosaurs were removed from this figure, and the clade Ornithocheiroidea was simplified, in order to make the figure concise.  

damage at its anterior edge, and therefore is more difficult to precisely determine the rostrum shape, but it seems to exhibit a rounded-triangular anterior rostrum shape (Fig. 6B). ML 2554 also shares in common a constriction of the rostrum just posterior to the spatulated end with the mandible of *G. macrurus* (*Howse & Milner, 1995*), a demarcation of the spatula which is more pronounced than in the other afore-mentioned gnathosaurines. This overall variability in rostrum shape and dental apparatus is likely linked to contrasting functional feeding movements, and perhaps even prey differences (*Ősi, 2011*; *Kellner et al., 2019b*), just as modern spoonbills today also vary in bill shape (*e.g.*, the Yellowbilled Spoonbill *Platalea flavipes* has a longer bill and narrower spoon than the Royal Spoonbill *Platalea regia*, with both occupying the same habitat but consuming different prey (*Hancock, Kushlan & Kahl, 1992*)).

Other defining features of the Ctenochasmatidae pertain to their dentition; namely, 25 or more teeth per side (with seven or more teeth in the premaxilla), the most rostrally-situated teeth being elongate with cylindrical crowns that project laterally from the dental border (*Unwin, 2003*). ML 2554 exhibits at least sixteen preserved teeth per side, with at least eight preserved in the spatula (damage impeding exact further determinations), making for an assignation therein, and making it further comparable with the tooth distribution of *G. macrurus* and *T. luciae*. Furthermore, if calculations of maximum tooth density are made based on the preserved portion, we can infer approximately 1.3 teeth per centimeter, making ML 2554 the gnathosaurine with the least dense dentition known so far. Tooth density is also informative in the Gnathosaurinae

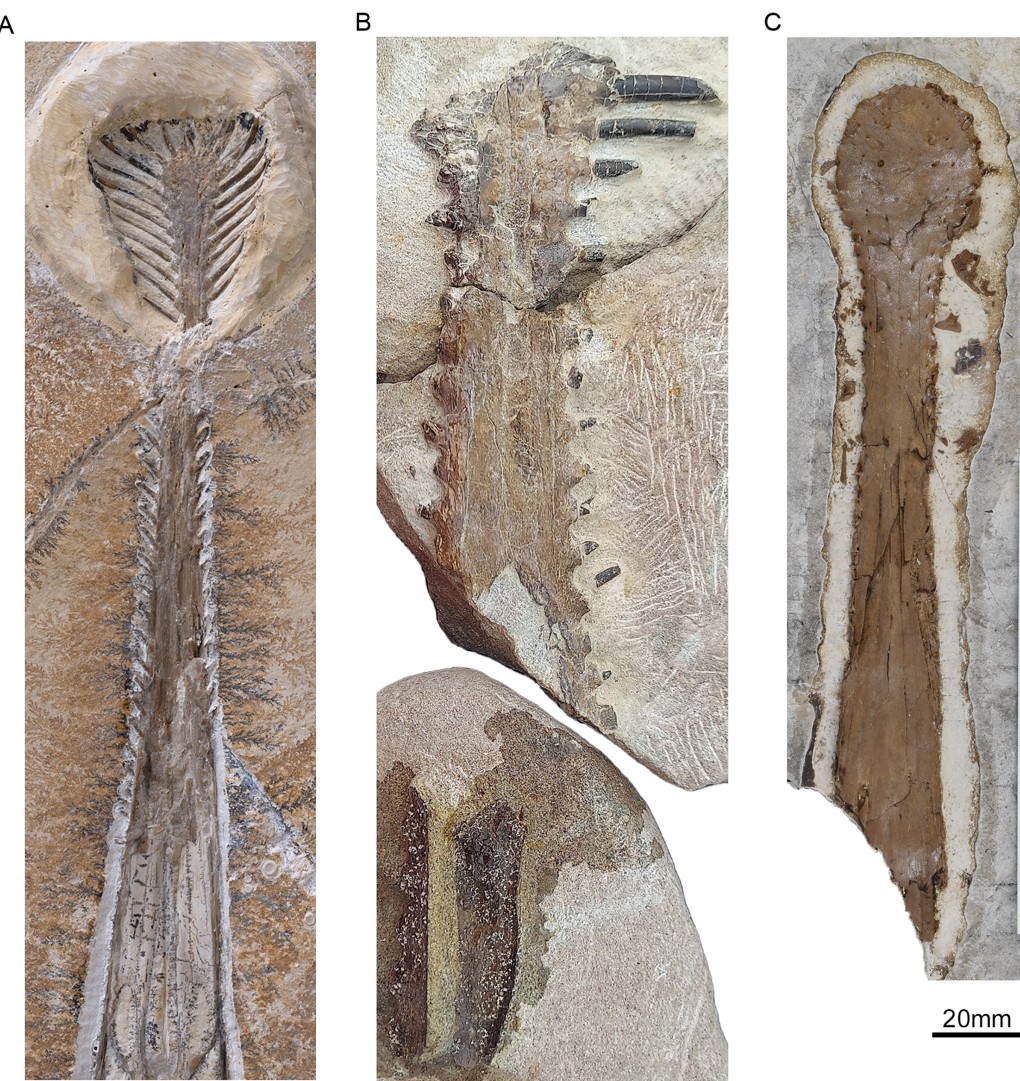

**Figure 6 Comparison of the ventral rostra of:** (A) *Gnathosaurus subulatus* Meyer, 1833 (JME-SOS 4580); (B) *Lusognathus almadrava* n. gen. et sp. (ML 2554), and (C) *Plataleorhynchus streptophorodon Howse & Milner, 1995* (BMNH R 11957).

(*Soto et al., 2021*), and even often autapomorphic within the Ctenochasmatidae, given that it can also potentially differentiate feeding specializations.

As in ctenochasmatids as a whole, gnathosaurines have conspicuous teeth set in alveoli, but gnathosaurines differ in having a raised collar of bone around their tooth bases (*Perea et al., 2018*), which is much more pronounced than in other ctenochasmatids, a feature well exhibited by ML 2554. This feature also contributes to the appearance of an inter-dental crenelation in ML 2554, what has been previously reported in *P. streptophorodon* (*Howse & Milner, 1995*), *G. subulatus*, and *T. luciae* (*Perea et al., 2018*; *Soto et al., 2021*). This collar also differs in being much more subtle than the distinctive alveolar parapet typically used to describe lonchodectids, which exhibit a more raised margin of the alveoli, usually

elevated above the medial portion of the occlusal surface (*Unwin, 2001*; *Averianov, 2020*), and which also have vertically-inclined teeth.

Similar to all ctenochasmatids, the teeth of ML 2554 show a sub-circular to ovoid cross-section, with smooth enamel. However, one significant difference of gnathosaurines from other ctenochasmatids is in their tooth robustness. Ctenochmastids are mostly filter-feeding, foraging in shallow waters with "needle-like" thin teeth, indicative of smaller prey items (*Wang et al., 2007*; *Bestwick et al., 2018*; *Paul, 2022*). The marked robustness of individual gnathosaur teeth suggests different feeding habits for their teeth being substantially larger and too widely-spaced for a true "filtration" functionality (*Bestwick et al., 2018*; *Knoll, 2000*), instead likely more associated to larger prey (*e.g.*, piscivory). This is not dissimilar to modern gavials, whose teeth are conical and slender, with pointed apices, the shape of which is widely associated with fish-piercing, since the pointed apex would facilitate penetration of prey items, and the slenderness of the teeth would enable teeth to slide between the bones of prey, minimizing breakage (*Massare, 1987*). Accordingly, the finer teeth of ctenochasmatids (other than gnathosaurines) would therefore be more utile in squashing or gripping prey items rather than for directly piercing them.

Another differentiation in the dentition of the Gnathosaurinae can be made in that their teeth taper overall, from base to tip (whereas ctenochasmatid teeth are only most apically tapered) (*Knoll, 2000*). Additionally, tooth projection itself can also be a relevant differentiator of feeding function, as the torsional stresses applied to teeth throughout prey capture and feeding also vary, depending on the nature and agility of that prey. Whereas procumbent dentition is found in *Rhamphorhynchus* sp. and also in most ctenochasmatids (*e.g.*, *Gegepterus changi*, *Ctenochasma elegans*, *G. subulatus*, and *T. luciae*), lateral projection is also found in certain rhamphorhynchids (*e.g.*, *Sericipterus wucaiwanensis*) and in ctenochasmatids (*P. streptophorodon*, *Liaodactylus primus*, *C. elegans* and *T. luciae*). ML 2554 has laterally-projected teeth in the spatular region, and also anterolaterally projected teeth posterior to the spatula (a feature that is found in other taxa such as *T. luciae* and *G. macrurus*), but different from taxa with posterolateral projections as inferred in *P. streptophorodon* (*Howse & Milner, 1995*). The recurvature of dentition could potentially aid in maneuvering prey back towards the gullet, as in modern gavials consuming fish (*Massare, 1987*), and because ontogeny could also potentially change tooth orientation (*Bennett, 2007*), this could potentially allow for feeding niches to change somewhat over the course of an animal's lifespan, and therefore this feature should be considered in a fluctuating capacity. Making any deductions on teeth alone should also be made prudently, since many taxonomic attributions for members of the clade are based on isolated teeth, and so may not be a wholly reliable metric.

Regarding vertebrae, the condition exhibited in one the partial vertebra of ML 2554 appears to follow the anteroposterior elongation typically found in both ctenochasmatids and azhdarchids (*e.g.*, *Howse, 1986*; *Martill, Sadaqah & Khoury, 1998*; *Unwin, 2003*; *Andres & Ji, 2008*), although its fragmentary state of preservation does not allow for any kind of insightful overall measurement. However, it is still informative in that there are no obvious

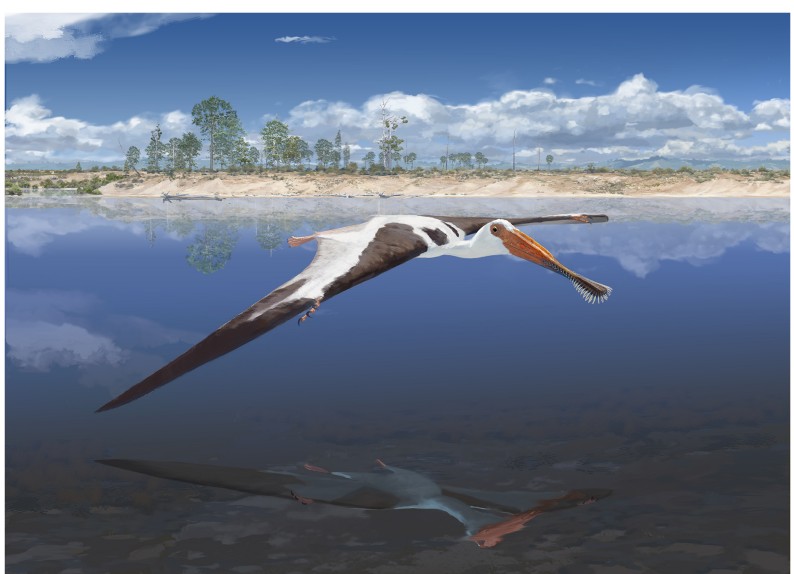

**Figure 7 Reconstruction of *Lusognathus almadrava* and its paleoenvironment by © Jason Brougham.**

sutures between the vertebral centrum and the neural arch, suggesting that it belonged to an osteologically-mature individual (*e.g.*, *Bennett, 1993*).

Portrayals of pterosaurs throughout the Triassic and Jurassic have traditionally been as relatively small animals, with wingspans constrained to around 1.6–1.8 m or less, whereas Cretaceous pterosaurs reached well above 3 m (*O'Sullivan, Martill & Groocock, 2013*; *Jagielska et al., 2022*). Of late, it has been increasingly postulated that Jurassic pterosaurs have been previously underestimated for their size range (and particularly when considering the larger end of the spectrum). Some previously-known larger-sized examples include the 1.73 minimum wingspan of *Sericipterus* (*Andres, Clark & Xing, 2010*), the 1.8 m wingspans of *Camplognathoides* (*Padian, 2008*) and *Rhamphorhynchus* (*Wellnhofer, 1975*), the 1.9 m wingspan of the rhamphorhynchine pterosaur from the Whitby Mudstone Formation (*O'Sullivan, Martill & Groocock, 2013*), and the 2.5 m wingspan of the Middle Jurassic sub-adult *Dearc sgiathanach Jagielska et al. (2022)*. There are also the proposed (but not wholly reliable) over 2.5 m wingspans of *Harpactognathus gentryii Carpenter et al. (2003)* (the fossil has been removed from museum collections since its publication) and the gnathosaurine *Huanhepterus quingyangensis Dong (1982)* (of unreliable age), the 3.3 m wingspan of a very fragmentary potential *Rhamphorhynchus* (*Spindler & Ifrim, 2021*), and the 3.5–5 m wingspan of a specimen described by *Meyer & Hunt (1999)*, but which may be scaled using an uncertain attribution, according to *O'Sullivan, Martill & Groocock (2013)*. Pterosaurs over the 2.5 m wingspan have therefore not typically been found to be older than the Early Cretaceous (*Meyer & Hunt, 1999*).

Notwithstanding these larger size estimates, the pterosaurs of the Jurassic of Portugal are especially remarkable for the time period (*i.e.*, the 4 m estimated wingspan of the fossil femur described by *Bertozzo et al., 2021*), with ML 2554 being no exception in corroborating the evidence of large Jurassic pterosaurs. The minimum overall length of the

preserved rostrum of *Lusognathus almadrava* begins at 20.2 cm (when all three rostral blocks are lined up) but could potentially be even larger. This means that if we scale the specimen with the overall skull dimensions of *Gnathosaurus subulatus* (following the skull/skeleton scaling by *Carpenter et al., 2003*), where the calculated total minimum skull length of ML 2554 would be 60.8 cm, then *Lusognathus almadrava* would have achieved a minimum wingspan starting at 3.6 m. Additionally, the coeval pterosaur track site of Peralta, less than 10 km to the south of Praia do Caniçal, records about 400 *Pteraichnus*-like tracks of different sizes (half manus/half pes). The pes track length varies from 5.5 to 15 cm, which indicates the occurrence of very large pterosaurs, preliminarily making *Lusognathus almadrava* a likely candidate for the trackmaker. Based on ML 2554, *Lusognathus almadrava* could potentially reach these larger sizes, making it one of the largest Jurassic pterosaurs known, and the largest known Jurassic gnathosaurine, having a size matched only by some Cretaceous records of the group.

## CONCLUSION

A new pterosaur taxon has been found from the Late Jurassic of Portugal, introduced here as *Lusognathus almadrava* gen et sp. nov. Several characters allow to ascribe this specimen to the Gnathosaurinae, namely its spatulate rostrum, robust comb-like dentition, and the pronounced rims of its tooth alveoli. The presence of this taxon in the fluvio-deltaic lagoonal environment that has been suggested to be representative of Lourinhã Formation in the Late Jurassic (Fig. 7) is in keeping with the modern analog of modern-day spoonbills, whose habitats generally include more shallow-water estuarine, tidal flat, coastal marshland, or even inland lake areas (areas where there are mud, clay, or fine-sand bottoms) with inflows of fresh, brackish, or salt water (*Hancock, Kushlan & Kahl, 1992*).

The paleobiological implications of such an exceptionally large-sized pterosaur in the Jurassic of Portugal denote and reinforce a thriving ecosystem, abundant with prey, in this case perhaps fish (as indicated by the robust teeth of ML 2554). Although it has been generally agreed upon that pterosaur body size steadily increased throughout and up to the end of the Cretaceous, ML 2554 adds more evidence for body sizes already having increased substantially by the end of the Late Jurassic (when compared with earlier forms); this growth having potentially been a response to filling a different ecological niche than their competitors, the birds (*Benson et al., 2014*; *Tennant et al., 2017*). Although transitioning to this larger size may have inadvertently contributed to their downfall later on (since fewer niches are available for larger animals, making big species more likely predisposed to decline (*Cardillo et al., 2005*)), and since larger terrestrial animals are sometimes disproportionately affected by extinctions (*Benson et al., 2014*), at least in the Late Jurassic paleoenvironments this extraordinary growth was potentially advantageous for their flourishing success.

## ACKNOWLEDGEMENTS

Immense gratitude is extended to Filipe Viera for his invaluable scientific discovery and donation of the fossil to the Museu da Lourinhã. Thank you to the Lourinhã Paleoteam for their good-humored willingness to excavate a partially-underwater specimen (in winter,

and at a moment's notice), with a special thank you to Alexandre Audigane and Miguel Moreno-Azanza for assisting in mobilizing the excavation permits (also in winter, and at a moment's notice). Heartfelt thanks are also extended to Carla Alexandra Tomás, Micael Martinho, Laura de Jorge, and Carla Hernandez for their impeccable preparation of the specimen, and to the rest of the staff of the Museu da Lourinhã. Thank you to the DinoParque da Lourinhã and the Museu da Lourinhã, and to Bruno Pereira and Maria Rios for being coordinators of the Departamento de Investigação do GEAL. Many thanks to Bruno Camillo Silva of the Sociedade de História Natural de Torres Vedras, Pedro Aquino, Flávio Domingues, and Tiago Ferreira at Micronsense Metrologia Industrial, Lda. for their invaluable CT scan implementation. Oliver Rauhut is thanked for contructive suggestions, as are Fabian Knoll, Dave Hone, Matías Soto, and a third anonymous reviewer, whose feedback greatly improved this manuscript.

### Funding

This work was funded by the Horacio Matues (PIIHM) Grant. Alexander W. A. Kellner received funding from the Fundação Carlos Chagas Filho de Amparo à Pesquisa do Rio de Janeiro (FAPERJ #E-26/201.095/2022) and the Conselho Nacional de Desenvolvimento Científico e Tecnológico (CNPq #313461/2018-0, #406779/2021-0, #406902/2022-4). Octavio Mateus was funded by the Fundação para Ciência e a Tecnologia, I. P. with National Funds from the Ministério da Ciência, Tecnologia e Ensino Superior. (Grant Numbers: UIDB/04035/2020, GeoBioTec). The funders had no role in study design, data collection and analysis, decision to publish, or preparation of the manuscript.

### Grant Disclosures

The following grant information was disclosed by the authors:
Horacio Matues (PIIHM) Grant.
Fundação Carlos Chagas Filho de Amparo à Pesquisa do Rio de Janeiro: FAPERJ #E-26/201.095/2022.
Conselho Nacional de Desenvolvimento Científico e Tecnológico: CNPq #313461/2018-0, #406779/2021-0, #406902/2022-4.
Fundação para Ciência e a Tecnologia, I. P. with National Funds from the Ministério da Ciência, Tecnologia e Ensino Superior.
GeoBioTec: UIDB/04035/2020.

### Competing Interests

The authors declare that they have no competing interests.

### Author Contributions

- Alexandra E. Fernandes conceived and designed the experiments, performed the experiments, prepared figures and/or tables, authored or reviewed drafts of the article, and approved the final draft.

- Victor Beccari performed the experiments, prepared figures and/or tables, authored or reviewed drafts of the article, and approved the final draft.
- Alexander W. A. Kellner analyzed the data, authored or reviewed drafts of the article, and approved the final draft.
- Octávio Mateus analyzed the data, authored or reviewed drafts of the article, and approved the final draft.

### Data Availability

The TNT Phylogenetic Data Matrix is available in the Supplemental File.

The data is available at Morphobank: Project 3968.

https://morphobank.org/index.php/Projects/ProjectOverview/project_id/3968

The CT data is available at MorphoSource:

https://doi.org/10.17602/M2/M530357.

https://doi.org/10.17602/M2/M530372.

https://doi.org/10.17602/M2/M530404.

https://doi.org/10.17602/M2/M530331.

### New Species Registration

The following information was supplied regarding the registration of a newly described species:

Publication LSID: urn:lsid:zoobank.org:pub:54B6D020-612A-4E23-ADEC-2CDE383B5C3B.

Lusognathus genus LSID: urn:lsid:zoobank.org:act:0681BFD9-A50C-4D5B-BC12-0E4EBA10E280.

Lusognathus almadrava species LSID: urn:lsid:zoobank.org:act:F36E8148-61FE-4CE8-AF4F-5F099364F817.

### Supplemental Information

Supplemental information for this article can be found online at http://dx.doi.org/10.7717/peerj.16048#supplemental-information.

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
