# Peer review of "A new gnathosaurine (Pterosauria, Archaeopterodactyloidea) from the Late Jurassic of Portugal"

_PeerJ, doi:10.7717/peerj.16048_

## Round 0.1 · original submission · Minor Revisions

Please, together with your unmarked revised manuscript, provide a marked-up copy as well as a document explaining how you have addressed the issues raised by the reviewers.

·

Basic reporting

Basically all good. I have various small comments in the marked-up document covering some issues but these are minor and easy to correct.

Experimental design

No problems with the description. The reporting on the phylogenetic analysis needs to be more detailed but from what is present there's no obvious issues.

Validity of the findings

Basically sound. There's a paragraph on tooth shape and orientation that I thin can be cleared up and some of the conclusions are basically off topic but again this fairly minor.

Additional comments

None about those mentioned above, this is a solid piece of work.

Reviewer 2 ·

Basic reporting

The paper is well constructed and includes appropriate literature. I do think that rather than all the measurements only being found in the supplementary material, the reader would benefit from the inclusion of a table showing key data, particularly information regarding measurement of tooth separation.

Experimental design

The research question for this paper is well designed as the specimen is clearly distinct and the authors make reference to appropriate data. I have included a suggested reference to include regarding some of the factors affecting preservational bias in pterosaurs.

Dean, C.D., Mannion, P.D. and Butler, R.J., 2016. Preservational bias controls the fossil record of pterosaurs. Palaeontology, 59(2), pp.225-247.

Validity of the findings

While the diagnosis and description are strong, as the tooth per cm measurement is not an absolute value, I would recommend the authors use a minimum rather than a median value. Since this measurement is part of the diagnosis and the animals large size and dental formula clearly suggest even a minimum value would be distinctive, I think that would be more valuable for comparative research later on.

Additional comments

One small part of the conclusion. The authors refer to the theory of bird competition driving size in Jurassic pterosaurs with clear certainty. However this theory has been questioned (many groups of animals experienced size increases after the Jurassic extinction event and pterosaurs do not appear to have directly competed with birds in most ecologies) and as the authors noted earlier relatively large pterosaurs, certainly larger than those used in the referenced studies, have been identified in Middle Jurassic and potentially Early Jurassic strata. I would recommend removing this line and limiting speculation on the positives or negatives of large size for pterosaurs in the Cretaceous as this reads as more speculative than intended.

Annotated reviews are not available for download in order to protect the identity of reviewers who chose to remain anonymous.

·

Basic reporting

The manuscript is concise and well written. However, I included a few suggestions in the revised pdf.

Experimental design

No comments

Validity of the findings

No comments

Additional comments

This manuscript is a valuable addition to the knowledge of Jurassic pterosaurs.
It increases the diversity of Gnathosaurinae. Also the combination of teeth orientations (spatular teeth laterally oriented, post spatular teeth anterolaterally oriented) is different to other known gnathosaurines. I look forward to the finding of a specimen with a complete spatula. Perhaps the spatula of Tacuadactylus luciae can be included in the last figure. Inferred size of the specimen is remarkable for a Jurassic taxon.

---

## Round 0.2 · accepted · Accept

I am now accepting your manuscript in order to expedite the process. However, there are still a few points for you to consider. Firstly, it remains unclear to me why you've indicated two years for 'Archaeopterodactyloidea Kellner' on line 208. Secondly, considering that the new generic name is derived from 'Lusitania' (Latin), the appropriate connecting vowel with 'gnathus' should be -i- rather than -o-. Thus, the genus name should be 'Lusignathus.' Lastly, I would suggest considering the option of acknowledging the reviewers, although the final decision is yours to make.

·

Basic reporting

All fine

Experimental design

All fine

Validity of the findings

All fine

Additional comments

None.

Reviewer 2 ·

Basic reporting

I am happy that the authors have taken on the reviewers comments and feedbacks, and made appropriate changes.

Experimental design

I am happy that the authors have taken on the reviewers comments and feedbacks, and made appropriate changes.

Validity of the findings

I am happy that the authors have taken on the reviewers comments and feedbacks, and made appropriate changes.

Additional comments

I am happy that the authors have taken on the reviewers comments and feedbacks, and made appropriate changes.